# Caffeine Boosts Weight-Lifting Performance in Rats: A Pilot Study

**DOI:** 10.3390/nu16132022

**Published:** 2024-06-26

**Authors:** Emanuel Pereira-Alves, Julia Machado-Pereira, Anibal Monteiro, Roberto Costa-Cordeiro, Vinod Chandran, Igor Jurisica, Eduardo Prado, L. C. Cameron

**Affiliations:** 1Lorraine Protein Biochemistry Group, Graduate Program in Neurology, Gaffrée e Guinle University Hospital, Rio de Janeiro 20270-004, RJ, Brazil; emanuelalves@edu.unirio.br (E.P.-A.); juliamachado@edu.unirio.br (J.M.-P.); robertoclaudioanatomia@gmail.com (R.C.-C.); 2Laboratory of Protein Biochemistry, The Federal University of the State of Rio de Janeiro, Rio de Janeiro 20290-250, RJ, Brazil; professoranibal@yahoo.com.br; 3Institute of Biological and Health Sciences, Federal University of Mato Grosso, Barra do Garças 78605-091, MG, Brazil; 4Arthritis Program, Schroeder Arthritis Institute, Krembil Research Institute, University Health Network, Toronto, ON M5T 0S8, Canada; vinod.chandran@uhn.ca; 5Division of Rheumatology, Department of Medicine, Institute of Medical Science, Department of Laboratory Medicine and Pathobiology, Faculty of Medicine, University of Toronto, Toronto, ON M5T 0S8, Canada; 6Osteoarthritis Research Program, Division of Orthopedic Surgery, Schroeder Arthritis Institute and Data Science Discovery Centre for Chronic Diseases, Krembil Research Institute, University Health Network, Toronto, ON M5T 0S8, Canada; juris@ai.utoronto.ca; 7Departments of Medical Biophysics and Computer Science and Faculty of Dentistry, University of Toronto, Toronto, ON M5G IL7, Canada; 8Institute of Neuroimmunology, Slovak Academy of Sciences, 845 10 Bratislava, Slovakia; 9Laboratory for Research in Physical Exercise and Metabolism, Federal University of Alagoas, Maceió 57072-900, AL, Brazil

**Keywords:** exercise, caffeine, weight lifting, animal

## Abstract

Caffeine is a well-described ergogenic aid used to enhance athletic performance. Using animal models can greatly increase our understanding of caffeine’s mechanisms in performance. Here, we adapted an animal weight-lifting exercise model to demonstrate caffeine’s ergogenic effect in rats. Male Wistar rats (315 ± 35 g) were randomly divided into two groups: one group received 5 mg·kg^−1^ of caffeine (0.5 mL; CEx; n = 5) and the other 0.9% NaCl (0.5 mL; PEx; n = 4) through an orogastric probe (gavage) one hour before exercise. Weight-lifting exercise sessions were performed over three subsequent days, and the number of complete squats performed was counted. Analyses of the area under the curve in all three experiments showed that the CEx group responded more to stimuli, performing more squats (1.7-, 2.0-, and 1.6-fold; *p* < 0.05) than the control group did. These three days’ data were analyzed to better understand the cumulative effect of this exercise, and a hyperbolic curve was fitted to these data. Data fitting from the caffeine-supplemented group, CEx, also showed larger S_max_ and K_d_ (2.3-fold and 1.6-fold, respectively) than the PEx group did. Our study demonstrated an acute ergogenic effect of caffeine in an animal weight-lifting exercise model for the first time, suggesting potential avenues for future research.

## 1. Introduction

Caffeine (1,3,7–trimethylxanthine) is among the most widely consumed psychoactive molecules. Xanthine is found in different plants’ seeds, leaves, and fruit, such as coffee, cocoa, mate, and guarana. It has been consumed for centuries worldwide in many dietary products, such as beverages, infusions, soft drinks, chocolate, and, more recently, energy drinks [1,2]. Coffee is the primary nutritional caffeine source in Western Europe and the United States, with about 89% of the adult US population consuming it daily [3]. In South America, in addition to coffee and carbonated soda, infusions using yerba mate (*Ilex paraguariensis* leaves and branches) are daily sources of caffeine for a broad population [4,5,6]. In Eastern societies, particularly in China and Southeast Asia, caffeine intake is also high, with an increasing trend in the last decades, mainly due to the high consumption of traditional infusions, such as black and green teas [6,7,8]. Still, xanthine ingestion might be underestimated, as sometimes caffeine is not declared on food or beverage labels [2].

In humans, caffeine absorption varies with gastric emptying, with a half-life of four to six hours and a peak of absorption of one hour [9,10,11]. Methylxanthine increases circulating catecholamines, raising blood pressure and heart rate [12]. Due to its hydrophobic character and poor albumin binding (10–30%), caffeine can easily pass through cellular membranes and the blood–brain barrier, freely entering and leaving tissues [13]. Caffeine’s ergogenic, motor coordination, and anti-fatigue effects are due to its action as an antagonist of adenosine A_2A_ receptors blocking adenosine on it, which causes more dopamine to be released in the striatum. This also suggests a significant impact on improving caffeine’s ergogenic effects in the forebrain [14,15,16]. Caffeine-induced stimulation provides performance benefits such as endurance, motivation, fatigue reduction, and increased mental alertness [8,17].

Human studies have used 3–6 mg·kg^−1^ body mass doses to achieve caffeine’s ergogenic role [9,10,11,18,19,20]. Acute caffeine ingestion promotes training-induced adaptations in power and speed. Previous bench press experiments indicated chronic pre-exercise caffeine ingestion improves velocity-load and power-load curves in upper limb exercises [21]. Caffeine intake increases the total amount of work performed during strength training sessions due to improvements in power production [21,22]. Caffeine also improves recovery between strenuous exercise sessions due to its hypoalgesic effect on delayed-onset muscle soreness [23,24]. Caffeine’s effects depend on doses and individual sensitivity [9,10,18,25]. Caffeine habituation—a desensitization of caffeine’s stimulatory effects—may result in an increased amount of caffeine needed to have the same antagonist activity on receptors [18,25]. Although caffeine’s ergogenic effects do not entirely diminish in chronic conditions, regular repeated exposure increases tolerance [26].

Xanthine was added to the list of banned substances by the International Olympic Committee in 1984 and the World Anti-Doping Agency (WADA) in 2000, but removed from the list in 2004, which led to athletes’ increased consumption of caffeine [27,28,29]. Over the past few decades, use of caffeine-based ergogenic aids, particularly among athletes and physically active people, has become widespread, especially through energy drink brands’ sponsorships of events in different sports [30,31,32]. However, caffeine remains on a WADA substance-monitoring list [33]. Therefore, understanding caffeine’s effects on performance and exercise is crucial to various fields of science.

The administration of paraxanthine also promotes increased nitric oxide availability and vascular function [34], which are related to decline with age [35]. Furthermore, findings from a study carried out in mice indicate the administration of paraxanthine promotes significantly enhanced aerobic endurance and muscular hypertrophy [35].

Animal models can help comprehend caffeine’s tissue and cellular effects, and our group used animals under stress induced by different types of exercise to study general principles and compare results with those obtained in humans [36,37]. Different animal models have been developed to study exercise. Rats have been the most used animals due to their muscle response being similar to that of humans [38]. To date, few studies have evaluated strength enhancement in rats after acute caffeine ingestion compared to the large amount of data obtained during rats’ swimming or running. In this investigation, with the hypothesis that caffeine can enhance the performance of rats similarly to that of humans during weight-lifting exercise, we adapted an exercise model to assess the caffeine ergogenic effect in rats during weight-lifting exercise [39].

## 2. Materials and Methods

### 2.1. Animals

In an initial allocation, ten adult male Wistar rats (315 ± 35 g) were divided into two groups of five animals using GraphPad QuickCalcs (GraphPad Software LLC., Boston, MA, USA). However, one rat presented locomotion problems and was removed from the experiment. Thus, groups were divided as follows: control (PEx: n = 4) and caffeine (CEx: n = 5). The small sample size was defined considering that this was the first study evaluating caffeine’s ergogenic effect in a repeated-movement weight-lifting animal model. Therefore, the initial intention was to demonstrate primary evidence regarding this effect in our exercise protocol. A scientist not related to this study was asked to administer an orogastric probe (gavage) of 0.9% NaCl (0.5 mL; PEx) or 5 mg·kg^−1^ caffeine (0.5 mL; CEx) one hour before the exercise session in a separate room. He was the only individual who knew the animals’ identities and relationship with treatment.

Rats were kept in individual cages; each rat was marked with a permanent marker on its tail with the letter of its group and its respective number in that group. Rats were in an environment with a controlled temperature of 22 ± 2 °C, with a relative humidity of 40 ± 5%, receiving a cycle of 12 h of light and 12 h of darkness per day. They were fed balanced feed and water ad libitum. This study strictly followed the Normative Resolution MTCI #57/2022 of the Brazilian National Council for Animal Experimentation Control (CONCEA). This study was approved by the Ethics Committee in Research of Tiradentes University (UNIT/SE/031005).

### 2.2. The Squat Machine

The exercise was performed in a system adapted from Tamaki et al., which is described in detail elsewhere [39]. Specific modifications to the Tamaki system are described as follows:Using a platform-type grid electrode with conductor gel, the stimuli site was transferred from the tail to the rat’s hind paw, allowing more freedom of movement.We installed a control circuit to measure the complete squat movement.Red and yellow LEDs were added to the electric control system, allowing the most accurate protocol evaluation. A red LED indicated when a pulse was administered, and a yellow LED was connected to a sensor on the machine’s main lever (activated when the rat performed an entire squat) (Figure 1).

Exercise sessions were filmed (Appendix A). Two independent researchers analyzed the videos in slow motion.

Electrostimulation occurred every 2 s for 1 s (1 Hz) using a transcutaneous electrical nerve stimulator (TENS) Quark, Dualpex 961 (São Paulo, Brazil), which was previously calibrated by the National Institute of Metrology, Quality, and Technology (INMETRO, Duque de Caxias, Rio de Janeiro, Brazil). The short pulse duration promoted adequate stimulation, inducing rats to perform voluntary movements and avoiding tissue damage.

### 2.3. The Exercise Protocol

Rats were adapted to the squatting machine for a week. During adaptation, neither load nor electrical stimuli were used. After this period, rats were tested daily for three days to determine the ideal electrical charge and frequency for the stimulation (also without additional load other than the machine and animal weight). The one repetition maximum (1RM) test was performed to establish the individual load. Although some rats supported loads greater than 300% of their body mass, we set 80% of the load obtained on the 1RM test as the exercise load. On the fourth day, we performed 20 stimuli (one second on; two seconds off) intervals, according to the Tamaki et al. model, until exhaustion [39].

The rats followed the same protocol for three consecutive days.

### 2.4. Statistical Analysis

Data normality was analyzed using the Shapiro–Wilk test (*p* > 0.05). The average number of squats performed by rats of the same group after each stimulus (AVG) was calculated. The area under the curve (AUC) was calculated for AVG ± SEM after stimuli. AUCs were compared using the Student’s *t*-test for independent samples. We calculated Cohen’s d effect size (d) when using the *t*-test and r effect size for the Mann–Whitney test. For non-normal distributions, we used the Mann–Whitney test.

The cumulative mean of the three days’ AVGs was also calculated for each group. This value is mathematically defined by the successive sum of the corresponding three days’ mean AVGs after a determined stimulus, along with the previous ones, until the last stimulus.

A nonlinear regression equation was used to fit the curve of cumulative values as follows:(1)$$y=\frac{S_{max}\times x}{K_{d}+x}$$

$S_{max}$ is the maximum value of y, and $K_{d}$ is the number of stimuli necessary to reach half of $S_{max}$. Statistics and fitting were performed using GraphPad Prism 10.2.3 (GraphPad Software LLC.).

## 3. Results

Caffeine is a well-known ergogenic agent in long-term exercise. To achieve the effect of caffeine in an acute animal model of weight-lifting exercise, we measured the number of squats per stimulus in rats. During all three protocol days, the caffeine group (CEx) had a greater response to the stimuli than the control group (PEx) did (*p* < 0.05). The three calculated mean AUCs of the CEx group were consistently larger (60–106%) than those of the PEx group. Also, the CEx group exercised longer than the PEx group did on two of the three protocol days (Figure 2).

To better understand these data, data were plotted as the accumulated mean AVG of the three experiments against the number of stimuli. A hyperbola better fits these data with fewer variables and constraints. The S_max_ in the supplemented group (CEx) was 2.3 times bigger than that of the control (PEx). The K_d_ was also 1.6 times bigger in the caffeine group (Figure 3).

To highlight the fatigue resistance induced by caffeine, we compared the AVGs of CEx and PEx groups during the first and last five stimuli in the three experiments. The CEx group demonstrated a higher response to stimuli than the control group (PEx) did during the first five and last five stimuli (Figure 4).

## 4. Discussion

Caffeine is a well-studied ergogenic resource with a well-described and accepted enhancement of muscle endurance, strength, and power in different exercise models [29,40,41]. As a psychoactive drug, caffeine can also lead to an additional placebo effect in humans [42,43,44]. Here, we used a modified animal model to study caffeine in weight-lifting exercises, where the placebo interference was annulated. In addition, an animal model can permit different pharmacological or biochemical studies. Due to the acute use of caffeine (a single dose once a day) and this study’s duration (three days), there were no other measurable effects on the rat’s physiology (blood pressure, cardiac frequency, or sleeping).

Animal models are crucial for better understanding biological processes, including exercise. For methodological reasons, rodent exercise protocols are usually used in running or swimming studies, while studies based on strength training are rarely adopted. In the early 1990s, Tamaki et al. proposed an animal model analogous to human weight-lifting training, allowing a better understanding of molecular and cellular mechanisms related to weight-lifting exercise [39,45,46,47,48]. Here, we adapted Tamaki’s apparatus to study acute caffeine ingestion’s effect on exercise performance. We introduced two new control circuits, which allowed us to evaluate the performance by quantifying complete squats performed by rats.

Our model used operant conditioning to make rats perform repetitive movements voluntarily. We used a grid electrode to evoke discomfort in rats’ hind limbs, inducing rats to perform squats. As an electrical stimulation, TENS avoids involuntary muscle contraction [49]. Inducing voluntary muscular contraction is essential to mimicking human exercise practice and making more reliable inferences from animal models to human models.

Several studies have used animal models to analyze the molecular mechanisms underlying the caffeine ergogenic effect [14,15,16,50]. As we used a small sample size for this pilot study and only evaluated performance, we proposed a first-step investigation. We highlighted this weight-lifting model as a promising method to expand the understanding of caffeine mechanisms of action in future studies. Also, this weight-lifting model can be adapted to other species, such as mice, allowing the study of differences and similarities across species. This model can be used to understand the effects of weight-lifting exercise physiology under several clinical conditions and interventions in future studies [51,52,53].

In this study, caffeine improved the rats’ performance during weight-lifting exercises. We used the AUC as a summary measure to analyze performance through 20 stimuli. Comparing AUCs from three consecutive days, the caffeine-treated group responded more (1.7-, 2.0-, and 1.6-fold) to stimuli than the control group did (Figure 2). Also, our data show that the treated group exhibited a S_max_ ~2.3-fold bigger than that of the PEx group (Figure 3). Furthermore, the higher response of the CEx group during the first five and last five stimuli might demonstrate a consistently greater response to stimuli caused by caffeine during the exercise (Figure 4). Our data concurs with what was previously described: rats’ acute caffeine administration decreased fatigue during swimming or treadmill-running exercise protocols [50,54,55,56,57].

To the best of our knowledge, this is the first study demonstrating caffeine’s ergogenic effect in rats during a weight-lifting exercise with repeated movements. Although this is an exciting finding, more studies are needed to determine the cause of the increase in squat repetitions. Performance improvements precede muscle hypertrophy in animal models, as there is a synchronization between muscle fiber number and performance gains independent of alterations in muscle mass [45,58]. As the rats performed squats against the same weight during the three days and muscle hypertrophy was not assessed, we cannot elucidate for force gain or increased resistance. Previous studies have demonstrated opposite results after larger doses of caffeine (19.7 mg·kg^−1^; 15 mg·kg^−1^; and 6 mg·kg^−1^) with only increased strength or enhanced resistance in the grip of rats’ front paws while rats were pulled by the tail [14,15,59].

Absorption and bioavailability of caffeine are generally similar between humans, dogs, rabbits, rats, and mice, with interspecies differences in the route of metabolism and enzymes involved in this process (for a comprehensive study of similarities and differences among species, we strongly suggest reading Arnaud 2011 [60]). It was not our goal to compare or extrapolate the totality of results in rat models for humans, but to advocate that an animal model can be useful for understanding systemic caffeine effects. In this study, we used an intermediate caffeine dose (5 mg·kg^−1^) as described for humans (3–10 mg·kg^−1^), considering the same absorption window peak (one hour) [40,41,60].

It was widely discussed that one of caffeine’s mechanisms of action could be increased Ca^2+^ release from the sarcoplasmic reticulum (SR), enhancing muscular contraction [61,62,63], including an uneven effect on different muscular fibers [64,65]. These results were used by previous studies to support the more prominent effect of caffeine increasing repetitions in the absence of strength enhancement in humans [66]. It is crucial to say that most of these studies were performed under supraphysiological caffeine concentrations. Caffeine concentrations in vivo are in the 10–50 μM range, and it is feasible to say that caffeine does not directly affect muscle fibers at physiological concentrations [40]. Caffeine antagonizes adenosine receptors, which causes more dopamine to be released in the striatum. This also suggests a significant impact on improving caffeine’s ergogenic effects in the forebrain [14,15,16].

The Tamaki apparatus privileges rats’ lower limb muscles, mainly the ones predominantly composed of fast-twitch fibers, such as the extensor digitorum longus and plantaris [39,67]. During Tamaki’s experiment, these muscles presented bigger hypertrophy compared to muscles composed of slow-twitch fibers, such as the gastrocnemius and soleus, which presented discreet hypertrophy [39]. Our hypothesis is supported by previous studies that proposed that the caffeine ergogenic effect might not be related to enhancing Ca^2+^ release from SR, as it was described only when caffeine was in high and non-physiological concentrations [40,68]. Therefore, more studies are needed to understand the mechanisms underlying the weight-lifting performance enhancement found in our study. However, our study certainly showed that this adaptation of Tamaki’s apparatus can be useful for understanding the effects of exercise and weight-lifting training on physiology.

Different human studies and reviews showed a caffeine-enhancement effect on endurance and strength during several exercise protocols [66,69,70,71,72,73,74,75]. Also, some reviews proposed that caffeine is more likely to increase repetitions than enhance strength during human weight-lifting exercises [66,72,76]. Thus, further studies are required to determine our protocol’s cause of endurance gain.

## 5. Conclusions

Our study demonstrated the caffeine ergogenic effect in an animal model analogous to human weight-lifting exercise. Although this effect is well described for humans, this is the first study to demonstrate it in animals.

Our protocol can be useful in understanding the molecular mechanisms underlying weight-lifting exercise performance. Adapting this model to mice can also allow the study of strength exercise in other physiopathological models and help determine shared vs. unique mechanisms across species. Therefore, the animal model used in our study can expand the understanding of weight-lifting exercise physiology and exercise effects in physiopathology.

## 6. Study Limitations

This was a pilot study with a small number of animals. Investigating the effects of caffeine in an animal model seemed interesting, given that our group has been researching xanthine for nearly two decades. Our main goal was to enhance an existing proposed system with a new platform and electronics, thus making the experimentation easier to follow.

We now aim to adapt the model for larger studies using mice to investigate the effects of strength exercise on different diseases.

## Figures and Tables

**Figure 1 nutrients-16-02022-f001:**
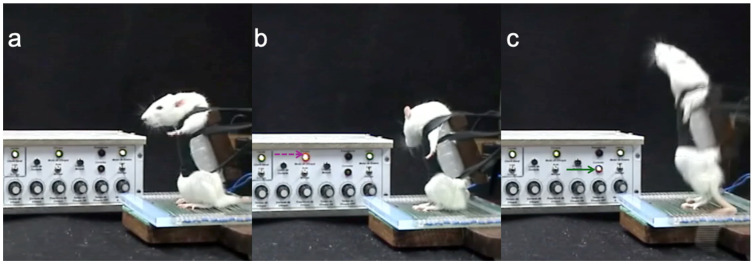
Three moments were captured from rat performance videos. (**a**) The rat is at rest, without stimulus. (**b**) Setting the stimulus, as indicated by the red LED (dashed magenta arrow). (**c**) The rat achieved complete movement after stimulation, as indicated by the flashing yellow LED (solid green arrow).

**Figure 2 nutrients-16-02022-f002:**
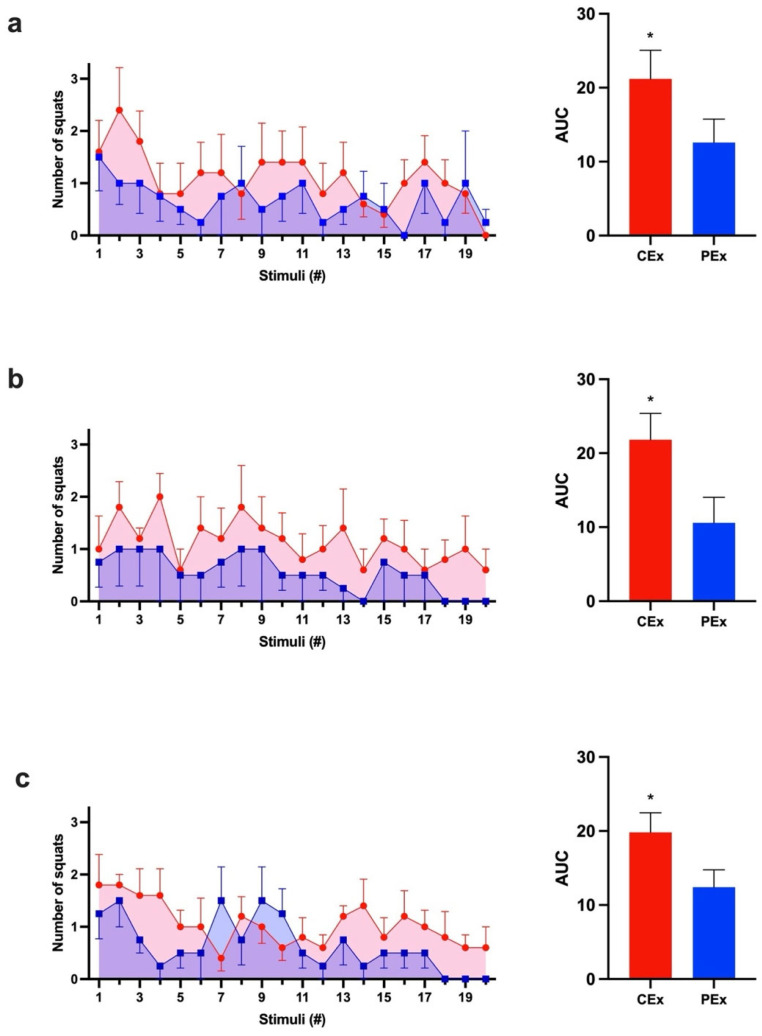
The caffeine group presented a greater response to stimuli during the three consecutive days. CEx (Red, ●); PEx (Blue, ■). (**a**) Day 1 graph of AVG ± SEM after each stimulus and Student’s *t*-test comparison between CEx AUC (21.2 ± 3.9) and PEx AUC (12.6 ± 3.2) (*p* < 0.05; d = 2.3; CI: 9.8 to 7.4). (**b**) Day 2 graph of AVG ± SEM after each stimulus and Student’s *t*-test comparison between CEx AUC (21.8 ± 3.6) and PEx AUC (10.6 ± 3.4) (*p* < 0.05; d = 3.1; CI: 12.4 to 10.0). (**c**) Day 3 graph of AVG ± SEM after each stimulus and Student’s *t*-test comparison between CEx AUC (19.8 ± 2.7) and PEx (12.4 ± 2.4) (*p* < 0.05; d = 2.9; CI: 8.2 to 6.5). * Significant (*p* < 0.05).

**Figure 3 nutrients-16-02022-f003:**
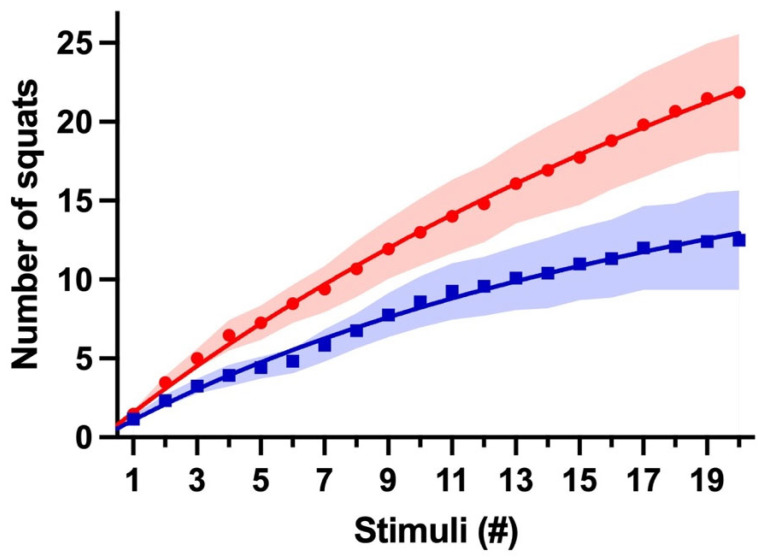
Caffeine-induced ergogenic response. This graph shows the cumulative mean AVGs of three days for CEx (Red, ●) and PEx (Blue, ■) groups. Lines represent CEx fitting hyperbola S_max_ = 69.5 and K_d_ = 43.2, and PEx fitting hyperbola S_max_ = 30.4 and K_d_ = 27.0; shading represents SEM variation.

**Figure 4 nutrients-16-02022-f004:**
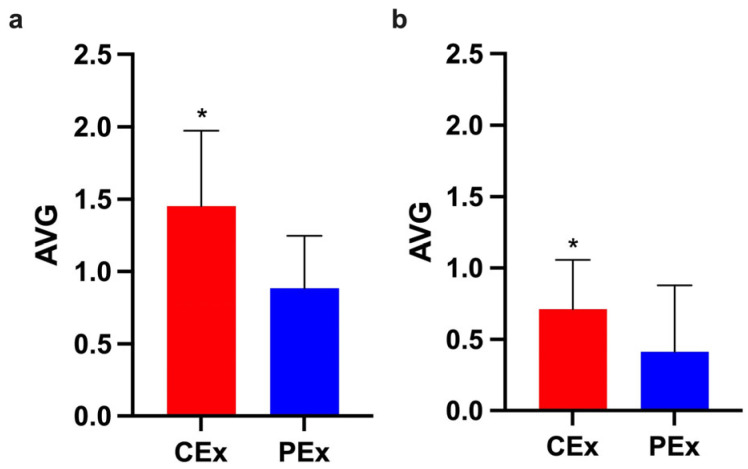
The CEx group had a higher response at the beginning and end of the exercise. These graphs present the mean ± SD of CEx (Red) and PEx (Blue) groups. (**a**) The CEx group demonstrated a greater response during the first five stimuli (*p* < 0.05; d = 1.3; CI: 0.2 to 0.9). (**b**) The CEx group demonstrated a greater response during the last five stimuli (*p* < 0.05; r = 0.4; U = 61.5). * Significant (*p* < 0.05).

## Data Availability

Original contributions presented in this study are included in the article; further inquiries can be directed to the corresponding authors.

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
