# Peer review of "Caffeine Boosts Weight-Lifting Performance in Rats: A Pilot Study"

_nutrients, 2024, doi:10.3390/nu16132022_

Round 1
Reviewer 1 Report
Comments and Suggestions for Authors
The manuscript of Pereira-Alves and colleagues is a carefully presented and beautiful piece of scientific research. Although the manuscript does not have any major scientific novelty, it provides a useful extension to rodents of a methodological framework to study the ergogenic effect of caffeine in the context of weight-lifting. In this sense, I warmly congratulate the authors for achieving this step forward.
The description of the methods and of the results is excellent.
1-I would only challenge the authors to consider repeating their quantifications focusing only on the first 5 stimuli and on the last 5 stimuli, which may enable to highlight a different effect of caffeine in relation to fatigue.
The main criticisms are related to the discussed molecular mechanisms underlying the ergogenic effects oof caffeine.
2-First, I do not accept as scientifically valid to claim that caffeine affects Ca dynamics in skeletal muscles, even if there are published papers claiming that it does. The quoted references (References 54-58) used as the lowest concentration of caffeine tested 1 mM, describing most of the effects at concentrations of caffeine of 4-15 mM. Animals are death by cardiovascular arrest when caffeine reaches sub-millimolar concentrations in the blood (accordingly no one has ever been able to measure concentrations of caffeine in humans higher than 28 micromolar in the blood or in any tissue). Hence the findings of these quoted references are totally devoid of any interest, either in term of physiology and even in terms of toxicology; for purely scientific reasons, they should not be quoted. And unless effects of caffeine at concentration of 10-50 uM (i.e. 500-times lower than the concentrations used in in vitro studies) are shown to occur in muscle fibers, it cannot be stated that physiological concentrations of caffeine have ANY effect on muscle fibers. These statements must be removed from the manuscript to avoid propagation of scientific mistakes.
3-The second aspect is closely related. The authors very rightly argue that the use of animal models is paramount to dissect the mechanisms underlying the ergogenic effects of caffeine. A Brazilian researcher from Santa Catarina did use animal models to show that the ergogenic effects of caffeine in running mice are critically dependent on adenosine A2A receptors (de Bem Alves et al., 2024, Front Pharmacol 15:1390187), namely of adenosine A2A receptors located in forebrain neurons (Aguiar et al., 2020, Sci Rep 10:13414), namely in the striatum (de Bem Alves et al., 2023, Purinergic Signal 19:673-683). None of these studies can be ignored and they should be quoted and discussed instead of the mambo-jambo of effects of caffeine at millimolar concentrations.
Reviewer 2 Report
Comments and Suggestions for Authors
Dear Authors
As one of the reviewers, I express my personal scientific opinion on your work. I would like to reassure you that I was trying to be positive and constructive but particularly as fair and honest as possible to your work. First of all, well done for the whole project and for the time spent to accomplish this task. I should also note that the work done on images and figures are positive points. However, the lack of the calculation of the Effect Size and CI and test-retest reliability, especially for the performance tests, are somewhat negative points.
Please accept my judgment with a positive and constructive way.
Introduction:
1. Very important and recent published (2020-2024) experimental articles are missing from the Introduction particularly and Discussion sections. There are only 3 referred articles, relevant to caffeine effect on exercise performance, which have been published during 2022 (two) and 2023 (one). The vast majority of the referred articles has been published prior to 2010, although there are several articles, relevant to the effect of caffeine on strength performance, which have been published during the last two years. This approach might question the development of the research question/hypothesis of the study.
2. In my point of view, your introduction should delve deeper specifically into the effect of caffeine on strength performance. For example, several studies (which some are referred into your manuscript) found already that caffeine indeed enhances strength performance in humans (i.e. see Giráldez-Costas et al. 2021; Guest et al. 2021 doi:10.1186/s12970-020-00383-4) and animals (i.e. Jäger et al. 2022, https://doi.org/10.3390/nu14040893). What is the novelty of your study and what new you have expected to observe?
Methods:
3. The research design is appropriate but the sample size is extremely small.
4. Lines 94-95: Since your sample size is obviously small and your initial intention was to demonstrate primary evidence regarding the effect of caffeine on strength performance have you ever considered in adding in your title the words …: “A pilot study”?
5. Have the animals been isolated individually or in groups? Please specify.
6. Which statistical test did you use to analyze your data?
Discussion:
7. You are extensively discussing the effect of caffeine on Ca2+. Why you did not evaluate Ca2+ release from SR?
Round 2
Reviewer 2 Report
Comments and Suggestions for Authors
Dear authors,
You have responded to my initial comments with professionalism and honesty. I am happy enough with all your responses to my initial comments and with the clarifications given and the amendments made in the revised manuscript. However, following my second review I have noticed that you have not reported any limitation of your study. I do personally believe that the study still has some limitations. Could you please report any potential limitation into your manuscript? In my point of view, your study could be published after incorporating the potential limitations in the revised manuscript.
I deeply apologize for not pointing out this to my initial review.
Best Regards
